# MyDiabetes—The Gamified Application for Diabetes Self-Management and Care

Nooralisa Mohd Tuah [1,*], Ainnecia Yoag [1] and Fatimah Ahmedy [2]

1 Faculty of Computing and Informatics, UMS, Kota Kinabalu, Sabah 88400, Malaysia; ainnecia@ums.edu.my
2 Faculty of Medicine & Health Sciences, UMS, Kota Kinabalu, Sabah 88400, Malaysia; fatimahmedy@ums.edu.my
* Correspondence: alisa.tuah@ums.edu.my

**Abstract:** Gamified applications are regarded as useful for patients in facilitating daily self-care management and the personalization of health monitoring. This paper reports the development of a gamified application by considering a design that had previously been investigated and reported. Numerous game elements were installed in the application, which covered several tasks aimed at managing diabetes mellitus. The development process utilized the Rapid Application Development (RAD) methodology in terms of system requirements, user design, construction, and cutover; this paper refers to the user design and cutover processes. The developed application was tested through system testing and usability testing. The usability testing adopted the Software Usability Scale (SUS) to assess the usability of the application. Twenty participants were involved in the testing. The result showed that the gamified application is easy and practical to use for an individual with or without diabetes. All the provided functions worked as designed and planned, and the participants accepted their usability. Overall, this study offers a promising result that could lead to real-life implementation.

**Keywords:** gamification; diabetes self-management; RAD methodology; game-design-based; Software Usability Scale

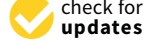

## 1. Introduction

Patients with long-term health conditions must adapt to a new routines and lifestyles, mainly involving their daily activities and dietary intake. Recent technology has revolutionized such activities by creating incredible tools and resources, and putting useful information at our fingertips. Despite the growing prevalence of smartphones, health-focused digital learning using the gamification approach has only been sparsely implemented in daily life. This omission may hinder an individual's efforts to self-care and manage, particularly those living with health conditions like diabetes mellitus.

Diabetes mellitus (DM) is one of the most common non-communicable diseases worldwide. It has become an important item on the agenda of healthcare providers. In recent years, researchers have focused on developing intervention tools that can foster self-care management. Unmanageable levels of blood glucose led to health complications and a decline in quality of life.

DM is an endocrine disease characterized by a person's blood glucose level [1,2]. A significant increase above the normal level will lead to a diagnosis of type 1 or type 2 diabetes mellitus. Healthcare services should promote the awareness of a healthy lifestyle and disseminate diabetic literacy and knowledge widely to prevent the condition from increasing. In Malaysia, DM has been increasing annually; a particular issue is the rise in Type 2 DM among Malaysian adults over 30 years old [2,3]. This increment is due to poor self-care management, minimal awareness of the disease, a lack of medication adherence, and dietary issues [2]. Individuals with DM require careful monitoring and care from the patients themselves, the primary care clinic, and the hospital. Currently, healthcare

providers utilize readiness questionnaires [2] to test individuals' knowledge and literacy of their condition, the related treatment, and the care required.

Meanwhile, game-related research scholars have recognized gamification as an approach that uses game mechanics in non-gaming contexts to facilitate healthcare management [4]. The gamification approach offers an attractive application that could encourage and help individuals understand and become informed about their condition. Gamification helps to balance learning and playful aspects while maintaining conditions of good health. In general, healthcare and gamification have been integrated and implemented in various diabetes applications. Gamification is seen as an approach with much potential in the healthcare field. Scholars have investigated gamification as a practical intervention tool for motivating individual engagement with treatment and care [4–6]. Moreover, for individual patients to learn about their condition through a playful environment is considered more convenient for them, and the prevalence of such environments will increase via mobile devices.

Research in gamification and healthcare has frequently been explored to provide viable options to patients and healthcare providers in facilitating diabetes self-care management. Previous research [7] showed no significant improvement in patients' medication adherence when using games for DM. Another study [8] has reported that using games for DM has no considerable effect on HbA1c reading. However, there was a significant result in terms of diabetes knowledge and awareness. Since then, researchers have studied and improved many aspects of this intervention, especially in the gamification and understanding of self-care. Previous research [1] has recommended the need for an effective platform, standardization, and simplification to ensure better knowledge acquisition.

Although the research on self-care management through fun learning games and gamification has been conducted, these studies only focus on a single diabetes issue, for example, problems of blood glucose [8], medication intake [7], and diet management [5]. Furthermore, the available applications (mention by, for example [9,10]) have existed in different platforms and environments, but some are quite expensive due to the need for external devices. In addition, more configurations are needed for these devices, leading to the disruption of the process of learning about self-care management. Additionally, diabetes self-management using mobile applications is seen as not sufficient, whereby the features of self-management in a mobile app are very limited and do not comprehensively cover the diabetes requirement [11,12]. Developing one platform that integrates several issues related to DM would enable a practical and convenient tool for knowledge acquisition and self-care management.

This paper presents the groundwork for developing a playful, integrated environment and gamification application for DM. This development stage is a continuation of the requirement stage that was reported previously in [6]. The development process is described, and the results from the system testing are presented. The developed gamified application in this paper is intended to contribute to enhancing a self-management and self-care platform with gamification and a fun learning environment. Most importantly, the report discusses users' access, pace, time commitment, and capability in terms of managing their own health condition.

This paper is organized as follows: first, this paper reviews works related to DM; then, it describes the materials and methods used in the study. The description of materials and methods includes the development methodologies, the gamified applications, and the testing procedures. The next section covers the presentation of the results and discussion based on the conducted system testing. Finally, the concluding remarks as well as recommendations for future work are presented in the last section.

## 2. Related Work

Following the framework of diabetes self-management by Al-Marshedi et al. [13], the component of self-management requires a logbook, data visualization, and trend alerts. A logbook can incorporate medication tracking, appointment tracking, and a log of every

task related to a patient's condition. The application summarizes these logs into a simple visualization chart/graph. The chart provides the resultant trends, with any unusual trends alerting the patient, carers, and doctors if the system is connected to them. Moreover, the self-management of diabetes is enhanced through educational games, apps monitoring, and in-game motivational feedback [9,10]. Included in the self-management functions of most mobile apps on the market, as reviewed by Priesterroth et al. [10], are the functions of a diary for insulin doses, and logs of food intake, activity, weight, and blood pressure. The reviewed articles researched these functions utilizing fun learning games and gamification over the years. However, those articles only focus on a single diabetes issue, for example, problems of blood glucose [8,9], medication intake [7], and diet management [5,14].

Research by Klaassen et al. [9] and Lewis et al. [15] use the PERGAMON framework, which utilizes wearable sensors, games, and gamification for their diabetes self-management application. The framework is designed as a gamification platform. It has many functions, such as a diary, mini-games, user profiles, and personalization. It uses a virtual coach, goals, and tasks as the main game elements, apart from points, badges, and levels. The game mainly involves empowering patients with the knowledge of controlling blood sugar levels through their dietary intake and carbohydrate counting tasks. Patients can also log any activities related to diabetes through the application, for example, exercise activities and daily water intake. Furthermore, the application allows the patients to create and customize their profiles. Nevertheless, these applications require external devices to be paired with the application.

In terms of the application of gamification for diabetes, Klaassen et al. [9] discussed the difficulty and complexity of diabetes games that are suitable for self-learning. They must be designed to be straightforward and simple. Moreover, the design of the games and applications must be aligned with the targeted users. Vassilakis et al. [16] further emphasized that in diabetes self-management, learning through digital applications or games is an alternative method used by healthcare personnel to support and motivate a patient to enhance their level of learning. In this view, giving feedback and guidance are essential, and ease of use should be considered in the design of applications. Other than using a gamified platform application, the literature reports that applications that are solely games have been utilized as tools to educate diabetes patients (such as [5,8,16]). Even though such applications have a specifically targeted diabetes solution, they are designed and built with multiple activities that facilitate the acquisition of the knowledge and skills related to diabetes. However, various aspects of diabetes knowledge and the scope of the related skills have yet to be discovered, and further research should be expanded to include diabetes-specific features. Vassilakis et al. [16] suggested that more games should be designed to educate patients in various aspects of diabetes, including the promotion of a healthy lifestyle, the need to avoid smoking, and encouragement to take up physical activities. Moreover, as reviewed by Priesterroth et al. [10], the gamified applications have frequently been implemented to promote self-management. However, gamification features are not systematically implemented in such applications.

Meanwhile, Brzan et al. [12], in their review of the 65 mobile apps for diabetes self-management, argued that the mobile apps development should employ user-centered design, which will specifically include individual patient's requirement in the application. The review also showed that only a small amount of content related to learning and self-management was embedded in mobile applications. The self-management indications used in [12] include monitoring blood glucose, nutrition intake, physical exercises, and body weight. However, it should not be limited to those functions only. Other self-management features suggested in [12], such as reminders, notes, tracking functions, and personal messaging, should be designed and applied in the mobile application. One particular issue in mobile apps implementation that should be considered is the feature of diabetes self-management in mobile apps is limited [11,12]. Huang et al. [11] advocate that many diabetes mobile apps lacked medication management features and had less emphasis on

basic reminder features. Thus, designing the self-management components is crucial and needs to carefully consider the requirement for diabetes and the requirement from patients.

Therefore, even though the available intervention tools utilizing gamification elements have been researched and invented, they represent different targets in improving a person's health condition. As self-management and care are the keys to health improvement, features related to these aspects must be designed accordingly. Thorough approaches to the design and implementation of gamification tailored to diabetes self-management requirements are needed to produce a more practical intervention tool.

## 3. Materials and Methods

This section focuses on the methodology and materials used in building the gamified application. A stepwise explanation of the processes involved in each phase is presented. Likewise, the materials employed are also described; those involved in this study are (1) the tools utilized in the development phase and (2) the gamified application itself. Moreover, this section outlines the gamification features and the implementation of the gamified application features.

### 3.1. RAD Methodology

The development of integrated playful games for diabetes mellitus follows the Rapid Application Development (RAD) methodology. RAD consists of 4 phases: requirement planning, user design, construction, and cutover. RAD is a method that focuses on system development through a prototype and reusable codes. Using the prototype, developers can obtain rapid responses from users during the development cycles. Meanwhile, the reusable codes from available open-source repositories enable the development period to be shortened. Moreover, this method accelerates the development period without impairing the quality of the application. Thus, RAD was chosen on the basis that developers and users work synchronously to create a functional product that follows the user's requirements. The RAD phases are presented in Figure 1.

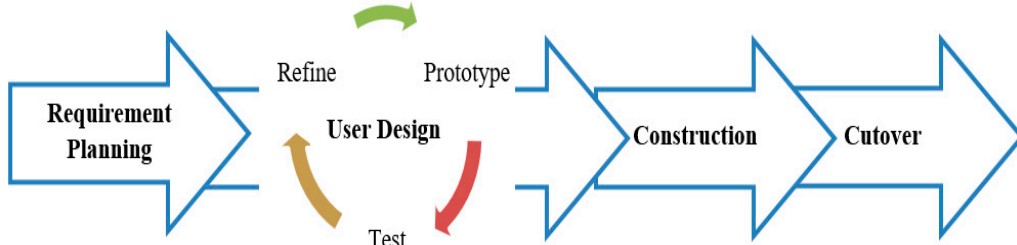

**Figure 1.** Rapid Application Development (RAD) methodology.

Based on the RAD methodology in Figure 1, each phase is described as follows:

(1) Requirement Planning phase: The requirements of a system are gathered and analyzed by the developer, after which a response from the users is received to confirm the requirement specifications. In this study, research previously conducted by the authors gathered the users' requirements through a user experiences approach. The authors implemented mock-up applications and storyboards to present the design ideas. This process involved a group of users from whom the researchers obtained their requirements through a focus group discussion. The results from the discussion were translated into a low-fidelity prototype, for which confirmation was sought from the users.

(2) User Design phase: The confirmed requirements from the previous phase are processed further by the designer or developers. Generally, this phase involves the iterative process of a detailed system design. Following the detailed design, a prototype is developed, tested, and refined based on the users' quick responses. In the context of this study, the authors created a high-fidelity prototype and obtained responses from the users to gain final confirmation of the design from the previous

phase. Then, the approved design was transformed into a complete system in the construction phase.

(3) Construction phase: Then, the high-fidelity prototype from the previous phase is improved into a fully developed system. In this phase, the essential aspects—the functions, interfaces, and databases—are integrated and completed. In the context of this study, the authors developed and tested the gamified application. The developed application was created with full system functionality, interfaces, and the integration of complete databases. Details of the game development approach are presented in Section 3.3, and the testing method is outlined in Section 3.5.

(4) Cutover phase: In this phase, the functions, interfaces, and databases are confirmed as a final system. In the context of this study, the authors conducted system testing to certify that the system worked as designed and required by the users.

Following the RAD methodology, phases 1 and 2 were successfully conducted and published, and this paper reports the work related to phases 3 and 4. The system's construction is presented in Section 3.3, and the system testing in the cutover phase is presented in Section 3.5.

### 3.2. The Development Tool

The diabetes gamified application in this study was developed using the PHP framework. The developers used the PHP code in coding the applications, the CSS for the interfaces, the JSON and JavaScript for the gamification element, and phpMyAdmin for the database. Hostinger.my was subscribed to as the server and hosting platform.

### 3.3. The Gamification Design and Development Approach

In developing the mini games, the game development approach by Hendrick [17] must be followed, which is a process that includes the prototype, pre-production, production, beta, and live. The prototype involves a process of translating the concepts into low-fidelity and high-fidelity designs. Pre-production involves the documentation of the game design. Production is the game development process, whereby the game assets, design, and code are constructed into a fully functional game. In beta, the game is tested to obtain feedback from the users. Once tested, the game is ready to go live. In this study, the process of prototyping was conducted in the user phase (stage 2) of the RAD methodology. The mini games planned for this study lay in the game production, beta, and live processes, which were the processes conducted in the construction phase (stage 3) of the RAD methodology. In the selection of the game elements and mechanics of the diabetes gamified application in this paper, two considerations were made.

First, the design is based on self-management elements in the gamification for the chronic illness framework in [13] and application of fun elements in motivating a person to sustain their engagement with a health-based gamified application [4,10,13]. With this in mind, the implemented game elements were a logbook (record-based), data visualization (graphs), and alerts. A logbook is any recorded data that relates to given features, such as data concerning medication, appointments, or tasks. For each type of data, the rate of completion is visualized in the form of percentages, using a circle graph on the user's dashboard. Alert messages pop up to remind the user when any of the data is reached or if the given due date has passed. Meanwhile, the selected fun elements are missions, the progression bar, avatar, and badges, as well as the challenges in the mini games. The element of missions in the gamified application allows users to set targets to improve their health condition. The achievement of the missions is visualized through the progression bar. Users will be intrigued to see their progress over time. When any mission is achieved through completing several tasks, a badge is awarded. This badge shows the specific achievement of the users, after which a different user status (novice, intermediate, advanced) is displayed on the user's account. This situation is anticipated to influence the users' engagement with and behavior toward better health self-management. The simulation model is illustrated in Figure 2. The simulation is using machination diagram.

As shown in the simulation, the sources are the data log from the users, in which the data are pooled according to its purpose and indicated by the progress mode. In the simulation, ten data pools were set to be achieved. Once completed, the data are pushed automatically (*) to another pool to indicate the data have been visualized on the application.

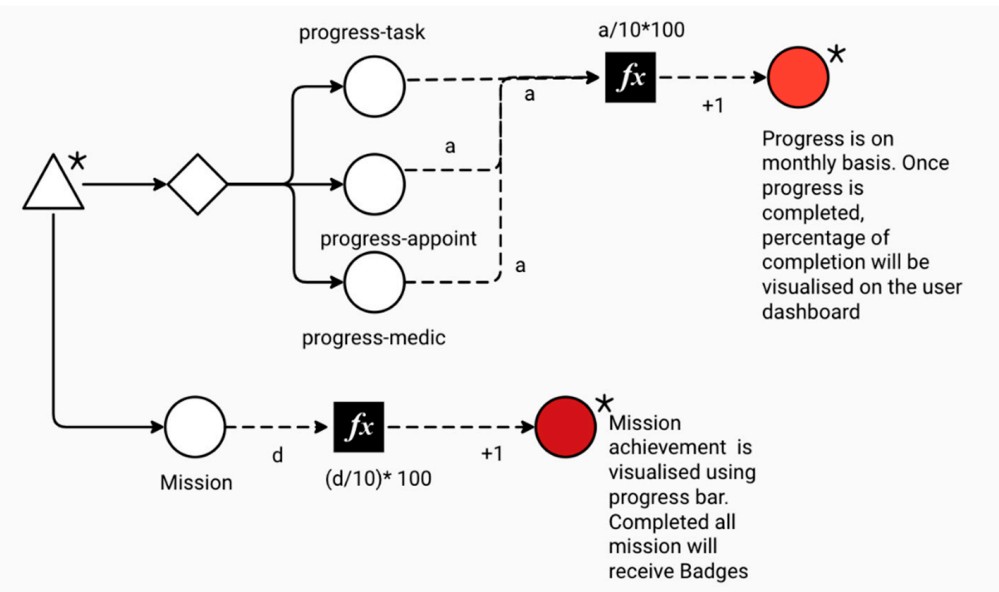

**Figure 2.** Model of the diabetes gamified application.

Second, the gamified application should follow a particular design pattern. In the literature, there are few available guidelines or frameworks that researchers or developers can use to assess the creation of the gamification design. For example, there is the Mechanic, Dynamics and Aesthetics (MDA) framework [18], Octalysis gamification framework [19], game design guideline [20], and the software engineering of gamification [21]. Each of the frameworks provides a different focus of how a researcher should develop a certain gamified application. However, each of them has a set of rules for good practices in gamification development and implementation. Among them, this study follows the game design guideline by Gallego-Durán [20]. The guideline was chosen because it helps the researcher design the gamified application design by analyzing the strength and weaknesses of the application according to the given characteristic. The guideline has ten characteristics of game-design-based gamification. The rubric of each characteristic is rated between point 0 (low), 1 (medium), and 2 (high). With that in mind, the gamified application yields the following scores:

- ■ (2) Open decision space. Users are in total control of the action taken in the gamified application (continuous space).
- ■ (1) Challenge. The mini games in the gamified application are composed of a series of levels with increased difficulty to challenge the user.
- ■ (2) Learning by trial and error. The mini games are instilled with features that enable users to keep on trying to gain knowledge related to their condition by playing the games. The users can play the games, complete the games regardless of the lost points, and live in the game.
- ■ (2) Progress assessment. The gamified application assesses the user's self-management activities' progress through graph visualization on the user's dashboard. Users who are progressing well and having good achievement will receive a badge.
- ■ (1) Feedback. Users receive feedback from the gamified application in the form of messages and reminders for incomplete tasks.

■ (1) Randomness. Some of the features are predicted. However, there will be a surprise movement in one of the mini games in which enemies will come out, and users need to avoid them.

■ (1) Discovery. A completed mission will unlock a new badge, new avatar selection, and new mission (health task) to accomplish.

■ (1) Emotional entailment. The mini games have a simple story and related character to target user emotion in learning about their condition.

■ (1) Playfulness enabled. Playing with the mini games may invoke playfulness with limited room for playing outside the rules set in the system.

■ (2) Automation. Even though users need to feed their data manually into the application, the progression, mission, badge, and achievement are automated.

For that, the gamified application gets a score of 14 points in total. The points show that the gamification design can plausibly be considered as accepted, as each of the characteristics is available in the application. However, the gamified application can still be added with more features in the future.

*3.4. The Diabetes Gamified Application*

The gamified application was designed by the developers (the authors) following the requirements collected in phases 1 and 2. The application interfaces were designed to be user-friendly. The application emphasized certain gamification features that were specifically designed for the users to take advantage of.

In the gamified diabetes application, several functions enable a person to manage their condition. The application requires a person to be registered. Once registered, they need to input and set the necessary information, such as their medication, appointments, tasks related to health targets, and other related treatment. Personal information and health-related data were also needed, for example, emergency contact details, physician details, allergies, and other co-morbid medical conditions. The application also implemented the concept of a personal dashboard, which was designed with the element of progression. This element shows the percentages completed monthly for each component. Visually displaying individual progress at a particular stage makes patients aware of their health status, particularly how well they are coping with their blood sugar control and current existing condition.

In the personal information feature, an element of badges and missions is included. A person receives a badge when he/she has completed or reached 100% on a particular component of the application. Meanwhile, the mission is another game element through which a person can track their health goals. This element of missions is also associated with the element of badges. For example, one individual health mission is to maintain their HbA1c reading at an average level in three consecutive months. From the recorded results, the application rewards the individual with a badge if he/she manages to achieve their mission. Another game element to be implemented in future designs is the element of points, which are received and collected from playing mini games. By playing a series of such games, a person can learn about their condition and obtain points, which can later be used to redeem rewards and items to customize their avatar. Figure 3 illustrates all the functions of the gamified application.

Based on the illustrated functions in Figure 3, the gamified application has three main sections: the user profile, self-management functions, and mini games. Users can manage their basic information via their user profile and add an emergency contact number and medical information (health condition) (refer to Figure 4). Self-management functions are presented through the dashboard (refer to Figure 5a). Users must manage their medication, appointments, and tasks related to their condition (refer to Figure 5b–d). Following the user design phase, three mini games will be installed, consisting of a memory game, an action game, and a role-playing game. The memory game involves memorizing matching pictures about food intake, the essential tools for diabetes, and healthy activity (refer to Figure 6a). Meanwhile, in the action game, users play an adventure activity in a given

environment in which they have to collect essential items for a person with diabetes (refer to Figure 6b). For the role-playing game, a rogue-like game will be installed.

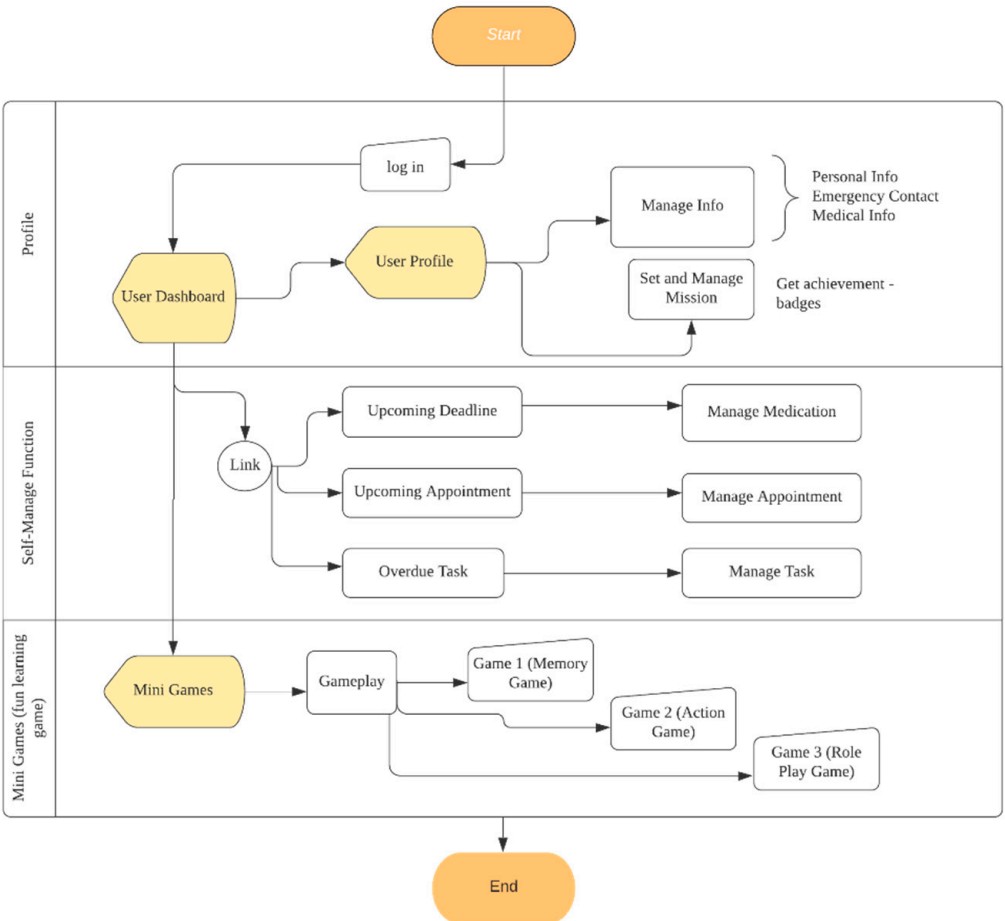

**Figure 3.** The functions flow in the gamified application.

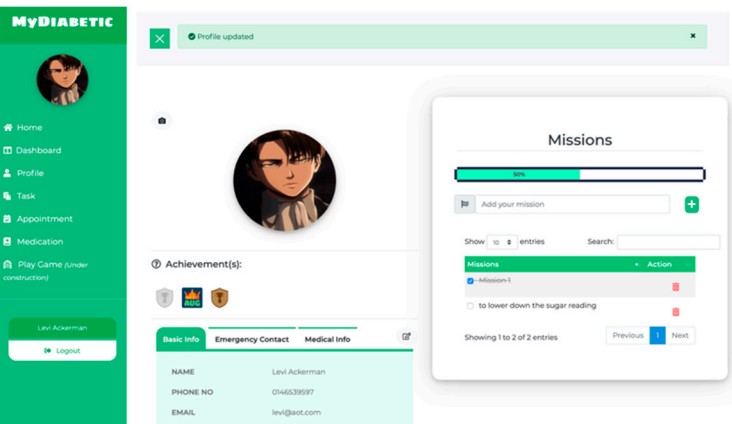

**Figure 4.** Gamified application—user profile, badge, and mission.

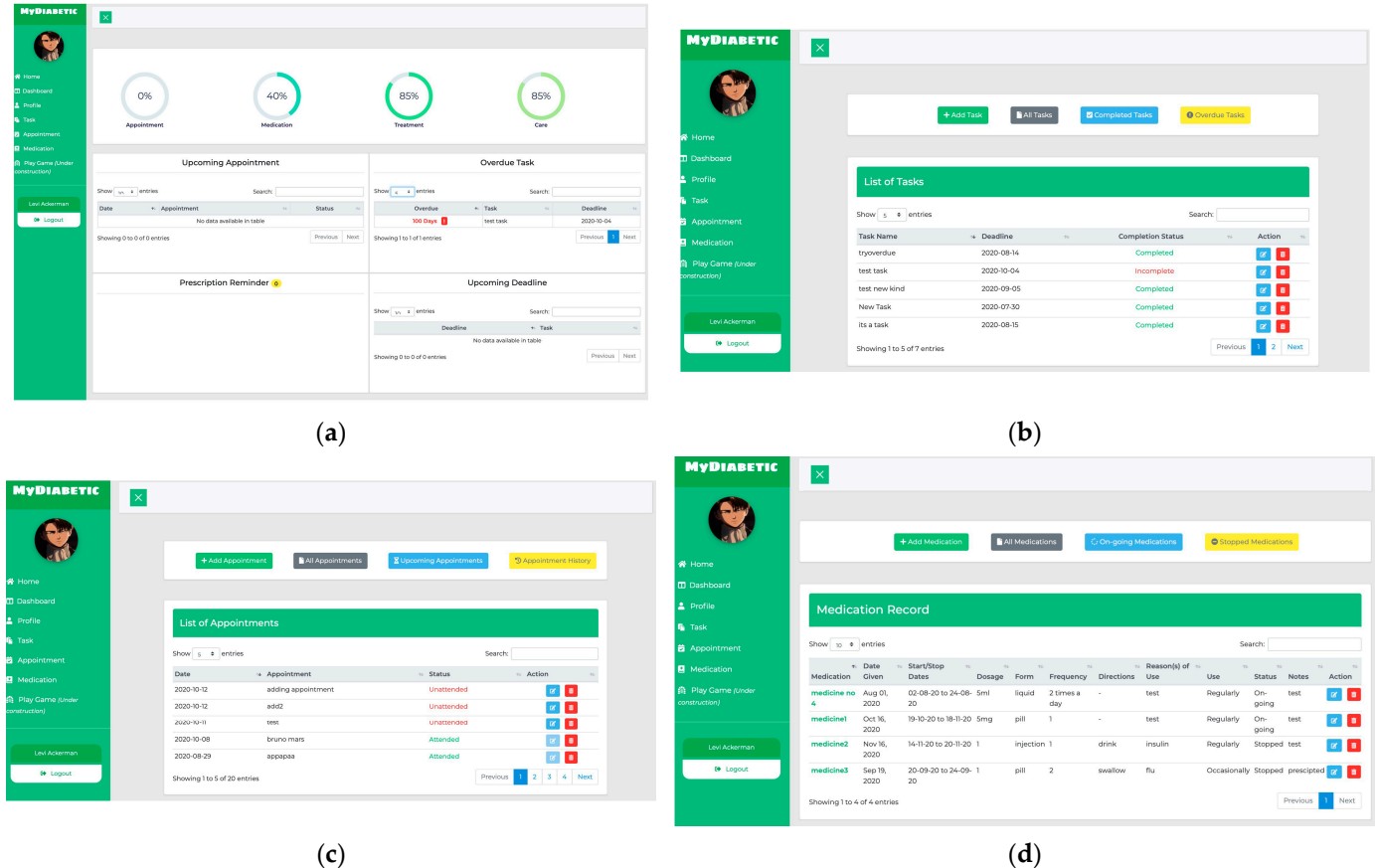

**Figure 5.** (**a**) Gamified application—Dashboard and progression; (**b**) Gamified application—Manage task; (**c**) Gamified application—Manage appointment; (**d**) Gamified application—Manage medication record.

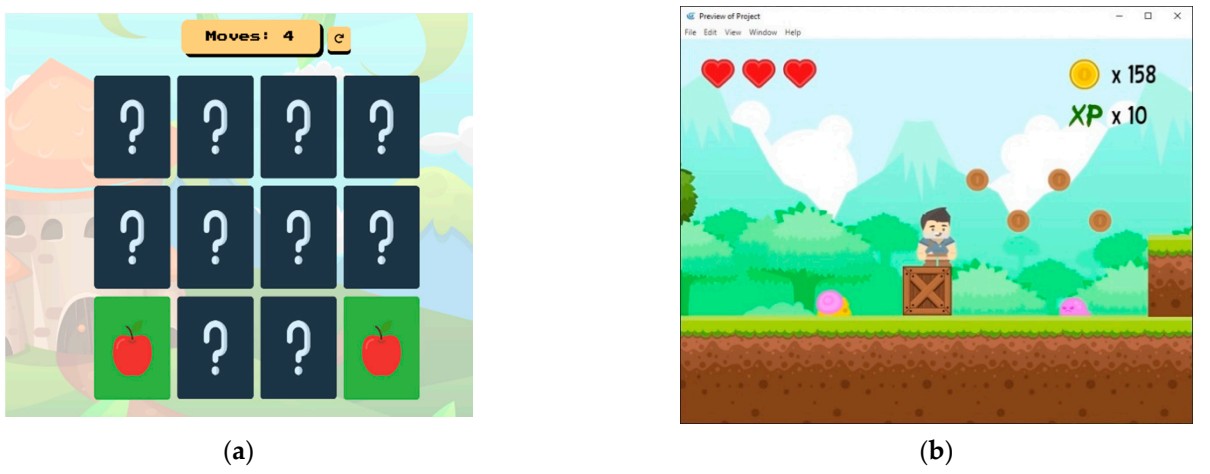

**Figure 6.** (**a**) Mini Games—Memory games; (**b**) Mini Games—adventure activity.

### 3.5. Software Testing Method

In this study, white box testing was conducted for each function in the gamified application. The testing begins with unit testing, which was followed by integration testing. Following that, once each module has been completely developed, the developer generates automated PHP testing. By generating the test, developers can ensure all units are programmed accordingly, and, more importantly, no errors have occurred in the application. The program codes can be identified as practical during the testing, thus minimizing the usage of computer memory resources during the operation (run time).

### 3.6. Empirical Research Method

Apart from the software automated test, preliminary user testing was also conducted to ensure the programs ran as designed and planned, and all transactions were successfully made without error. This testing was deemed necessary before the researcher could conduct acceptance testing with the potential users (diabetics). In this testing context, the errors were identified from users' misconceptions in determining the system flows. Feedback was also collected regarding the application interface and the way the system worked. For this purpose, the researcher utilized the established Software Usability Scale [22] to determine user perspectives from their use of the application. All the testing processes are illustrated in Figure 7.

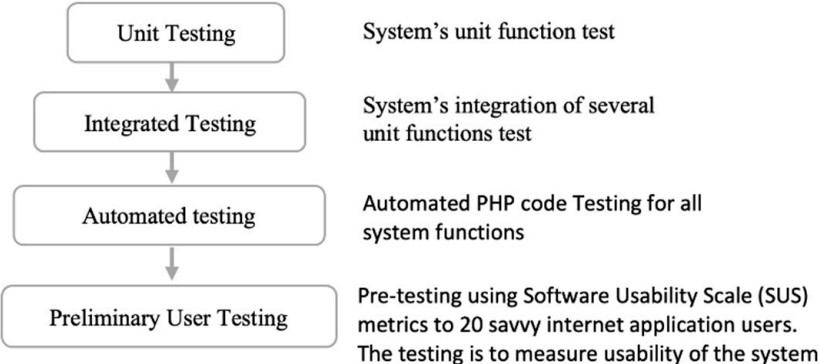

**Figure 7.** The process of the conducted testing.

Participants and Research Design

The study recruited 20 individuals for the preliminary user testing. These were randomly selected based on their level of familiarity with technology. They had to be familiar with online applications and have a higher internet usage in their day-to-day activities. This number of participants was considered plausible to enable the identification of a reasonable proportion of problems in heuristics usability [23]. Participation was voluntary, and no compensation was given for involvement. The participants were of mixed backgrounds and included persons with and without diabetes.

The testing was designed to be conducted by the users themselves. A call for participation was made via social media and the project website. Interested participants were randomly selected and formally emailed to gain their consent and provide them with participation details. Following the first email, a second email was sent to the participants giving detailed instructions about the testing. The instructions included the step-by-step process for conducting the test and the documentation needed for the testing. The testing was undertaken individually using online resources, and the testers had to refer to the given test cases and complete the testing within the allotted time.

## 4. Result and Discussion

Twenty individuals participated in the testing. The demographics of the participants are summarized in Table 1. Among the 20 participants, 16 were female and four were male. Their age distribution was between 30 and 40. Five of them had diabetes and 15 did not. However, their diabetes condition was controllable and not severe. These participants were also categorized as being familiar with technology and spent more than three hours per day doing online activities.

**Table 1.** Participant demographics.

| Demographic | No |
|:---:|:---:|
| **Gender** | |
| Female | 16 |
| Male | 4 |
| **Age** | |
| 30–35 | 12 |
| 35–40 | 8 |
| **Health Condition** | |
| With Diabetes | 5 |
| Without Diabetes | 15 |
| **Tech-savvy hours of daily online interaction** | |
| 3–5 | 5 |
| 6–9 | 12 |
| 10 and Above | 3 |

The results of the testing are presented according to the testing activities conducted. There are two parts, the automated system testing and the preliminary user testing. These results suggest the automated unit testing shows no significant errors. The application performance based on the server-side evaluation can be credibly interpreted as successful and suitable for the application environment. With ten simultaneous usages, only 1% of the server memory was utilized (out of 512 MB server memory) with an average response time of 0.335 per second. Based on the results, the application performance was manageable. Users could use the application widely with minimum delay, subject to their network and server performance. Table 2 shows the results of the application server-side performance.

**Table 2.** Application's server-side performance.

| Time (Min) | Load (# Users) | % Memory Utilization | Response Time (Secs) |
|:---:|:---:|:---:|:---:|
| 1 | 10 | 1 | 0.335 |

The system testing was conducted with 20 participants, and according to the test cases, all functions worked accordingly. The participants were able to follow the system's flow correctly. Thus, no system errors were found during the testing. Apart from conducting the test, these 20 participants also provided additional responses that reflected their opinions of the application. The responses were based on the given questionnaire adopted from the Software Usability Scale (SUS). The questionnaire used a Likert scale of 1 (Strongly Disagree) to 5 (Strongly Agree). There were ten questions, and the results of the mean and standard deviation (SD) of each question are shown in Table 3.

**Table 3.** Mean and SD of users' responses.

| No. | Questions | Mean | SD |
|:---:|:---:|:---:|:---:|
| 1. | I think that I would like to use this system frequently | 4.50 | 0.69 |
| 2. | I found the system unnecessarily complex | 2.15 | 0.37 |
| 3. | I thought the system was easy to use | 4.90 | 0.31 |
| 4. | I think that I would need the support of a technical person to be able to use this system | 1.35 | 0.49 |
| 5. | I found the various functions in this system were well integrated | 4.05 | 0.22 |
| 6. | I thought there was too much inconsistency in this system | 1.20 | 0.41 |
| 7. | I would imagine that most people would learn to use this system very quickly | 4.00 | 0.46 |
| 8. | I found the system very awkward to use | 1.65 | 0.75 |
| 9. | I felt very confident using the system | 4.80 | 0.41 |
| 10. | I needed to learn a lot of things before I could get going with this system | 2.15 | 0.37 |

The perceived usability of the gamified application was found to be highly reliable (10 items, $\alpha = 0.98$). Based on the results of the responses, as shown in Table 3, the positive questions (Q1, Q3, Q5, Q7, Q9) received a mean value of 4.0 and above. Furthermore, the negative questions (Q2, Q4, Q6, Q8, Q10) received a mean value of 2.5 and below. Additionally, based on the SUS scores interpretation, the total scores for each participant were multiplied by 2.5 to convert the scores into a 0–100 range. Scores above 68 were considered above average, indicating acceptable usability. Users rated the gamified application as very positive, with an average score of 76.87; the obtained score was above the average SUS score. The obtained SUS score recorded a median score of 76.25; the minimum score was 70; and the maximum score was 85. Figure 8 shows the boxplot of the SUS scores for all users.

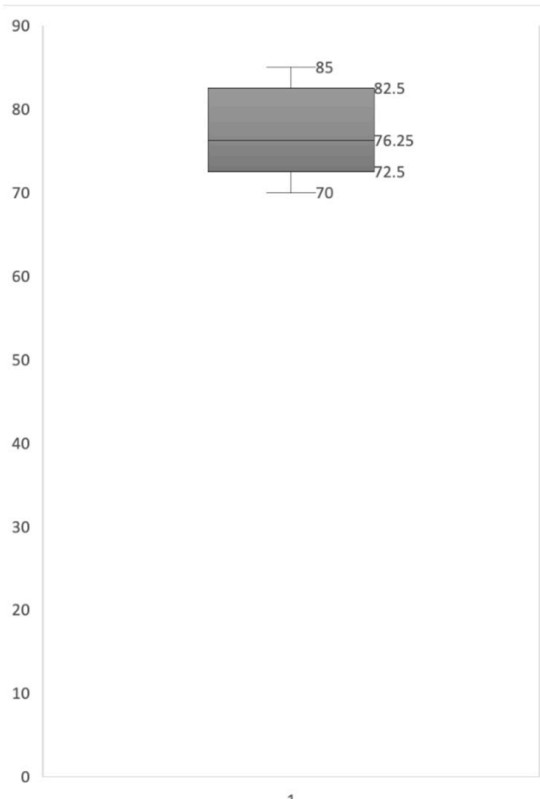

**Figure 8.** Software Usability Scale (SUS) overall participants scores.

The results in Table 3 and Figure 8 show that all participants agreed with the functions provided in the application, that the application was not complicated, and that learning to use the application was easy without the need of a technical support person. Thus, the users gained a reasonable level of confidence in using the gamified application. The users understood the process and were willing to use it further. Therefore, we assumed that the application was ready for actual user acceptance testing (tested by a diabetes patient). Although a direct comparison with the previous study in [10] may not be applicable, due to the different focus of the diabetes self-management implementation, the findings show that the gamified application implemented in this study is systematic, with consistent but not complex features. A comparison with the outcome of previous studies in [9,24] reveals a similar pattern in the need for a simple application whose functions are not overly difficult and whose design is relevant in its gamification elements and techniques. Hence, this finding has established the underpinning concepts of applying gamification for health self-management.

Nevertheless, certain limitations of this study were noted. First, the conducted testing was a self-regulated activity, which was conducted online due to the pandemic situation and movement restriction order. This resulted in a limited level of observation of user

behavior during the activity. Moreover, the testing activity was conducted over a short period. Thus, future work should consider longer experimental periods and evaluate a person's improvement in their health condition when the application is used. Second, one mini-game (the role-play) still needs further improvement, as it received several comments during the testing. Most comments concerned the patient's avatar (role-play character). It was suggested that the avatar should reflect the level of a patient's condition in the gameplay and gradually change as the condition of the patient improves. This suggestion is in line with the avatar implementation in a previous study [5] in which the avatar changes further explain the positive effect on the user's engagement in the gameplay. Meanwhile, other comments were directed toward the interface designs, which have been altered by the developers. Nevertheless, any comments and suggestions on the current functionalities could inform further improvement.

### 5. Conclusions and Future Work

The application of gamification for diabetes mellitus is gradually receiving attention as a tool and part of an individual's daily life activities. Providing an application that could help individuals learn more about their health condition indirectly teaches and encourages them to self-care and self-manage. Providing such an application also allows individuals with diabetes to adapt to their daily routine by themselves. However, individuals with little or no familiarity with using the Internet and technology find such applications challenging to use. This scenario could occur with older adults who are more accustomed to manual book records, nurse call reminders, and the physical diabetes awareness program. Nevertheless, personalized healthcare monitoring, such as the developed gamified application presented in this study, has been created for anyone who requires assistive tools in self-managing their diabetes condition.

This research reports the development and testing work related to the completion of a gamified application. The work was grounded in the RAD methodology, with the requirement and design phases having been completed. The developed application underwent preliminary user testing to assess the application's usability with the Software Usability Scale (SUS), and the results were encouraging. The results from the usability study show that the gamified application is generally easy and practical to use whether the individual is living with or without diabetes. The users also indicated that they would like to use the application frequently. However, currently, there is no proof that the system could improve a person's health condition. This should be taken into consideration in future studies.

Therefore, in future work, the researchers will conduct acceptance testing and assess the application's effectiveness for prospective users. A longitudinal study inspecting how a person could benefit from the gamified application, as well as how the application could affect the condition of a person's diabetes, will be further researched. The longitudinal study is considered necessary to measure any medical impact on a person when using a particular system application. The system effectiveness requires time and a monitoring method, such as a diary, to acquire comprehensive results. In the interim, suggestions for application improvements will be put into action. Meanwhile, other application improvements, such as developing a mobile apps version and adding more mini games, will be considered for future work.

Generally, the developed gamified application in this study can be considered a possible future solution for modern healthcare services. The application is an open platform, which currently involves diabetes as the subject of interest. Applying other health conditions as subjects of a gamified application can also be further explored.

**Supplementary Materials:** The following are available online at https://www.mdpi.com/article/10.3390/computers10040050/s1.

**Author Contributions:** Conceptualization, N.M.T. and A.Y.; methodology, A.Y.; formal analysis, F.A.; writing—original draft preparation, N.M.T.; writing—review and editing, A.Y. and F.A. All authors have read and agreed to the published version of the manuscript.

**Funding:** This research is funded by the Universiti Malaysia Sabah, grant number SLB0201-2019 and the APC is funded by the Universiti Malaysia Sabah.

**Institutional Review Board Statement:** The study was conducted according to the guidelines of the Declaration of Helsinki, and approved by the Medical Research & Ethics Committee of Ministry of Health Malaysia (NMRR-19-1732-49011 and 7 October 2019).

**Informed Consent Statement:** Informed consent was obtained from all subjects involved in the study.

**Data Availability Statement:** Data are contained within the article and can be found in the Supplementary Materials.

**Acknowledgments:** The authors would like to thank Universiti Malaysia Sabah for this research opportunity and the financial support during the completion of this research. In addition, the authors would like to thank the research assistant who administered and coordinated the testing accordingly.

**Conflicts of Interest:** The funders had no role in the design of the study; in the collection, analyses, or interpretation of data; in the writing of the manuscript, or in the decision to publish the results.

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
