# Peer review of "MyDiabetes—The Gamified Application for Diabetes Self-Management and Care"

_computers, doi:10.3390/computers10040050_

Round 1

Reviewer 1 Report

Very interesting and correctly written manuscript, with appropriate structure, research methodology and results.

There are few minor corrections to be made, in aim to have a better organization of the manuscript:

1st correction - Introduction is too long. It should be separated into 2 sections:

1. Introduction - presenting research motivation, basic terms and the rest of the paper structure

2. Releated work - presenting previous research from other authors.

2nd correction - subsection Testing should be renamed into "Software testing methods" and empirical research methodology or to separate it into 2 sections: "Software testing methods" and "Empirical research methodology"

3rd correction - authors should provide more detail on the methodology of experimental research sample creation, i.e. to explan how did they find persons to be included in the user testing of the software. What method did they use for the selection of candidates for the empirical research? Authors should emphasize are these persons Diabetes patients and were there any users that are not Diabetes patients? 

4th correction - in the conclusion part, there should be more explanation of how the system could improve the patients' health condition. The usefulness of the system has been measured with general usability measurement system, but the true usefulness should be directed towards medical condition of the patients. Surely, it is not easy to have measured the medical impact of using the software system, but the authors should think about it in their future work. Therefore, the conclusion should be extended with more possible directions in software improvement, but also in empirical research and scientific contributions that could be made in future, as continuation to this research.

Author Response

Dear Reviewer,

Thank you for the given comments. We have addressed the comments and suggestions as in the attachment.

Thank you.

Reviewer 2 Report

The study describes the development of a gamification application for diabetes self-management. The results of the evaluation with 20 participants using the Software Usability Scale (SUS) are presented.

Comments:

  1. In the abstract, add 1-2 sentences about the main results of usability evaluation. Was a developed application acceptable for users?
  2. What is the novelty and contribution of the developed applications with respect to many other similar applications available? Clearly state it at the end of the first section.
  3. Several similar applications are discussed in the introduction section, however, what motivated you to develop yet another application. Discuss the limitations of previous work as a motivation of your work.
  4. The development of the application was based on Rapid Application Development methodology, which is a general tool for developing any kind of application. Since you aimed to develop a gamified application/game, what specific game/gamification development methods were used? Did you use the patterns of gamification? (see, “Gamification of a project management system”). While the game is the focus, the article does not explain the game scenario and gamification techniques employed to support player engagement and motivation to play. These concerns should be discussed in more detail.
  5. Describe game mechanics of serious game using the Machinations diagrams https://machinations.io/
  6. Explain how you maintain user interest in continuing using this application. What specific elements of gamification (badges, etc.) did you use to retain user motivation and interest in continuing using this application?
  7. Figure 7: replace with a boxplot. Explain the results in more detail.
  8. Present a critical discussion on the limitations of the serious game for medical purposes / healthy lifestyle as well as any threats to the validity of the results. Make a connection between your results and what were the results of previous similar studies, and the new and unique thing that you created. Also explicitly say what we can learn from your work.
  9. Conclusions should be improved as it lacks insights and recommendations for further research in this domain. Use the SUS questionnaire results to support the claims on the attractiveness of the game.
  10. The references are not cited in the order of appearance as the journal requires.
  11. The language should be improved.

Author Response

Dear Reviewer,

Thank you for the given comments. We tried to address all the comments and suggestions. Please refer in the attachment for your further perusal.

Thank you.

Round 2

Reviewer 1 Report

Interesting software and an initial empirical research related to the attitudes of users which are both Diabetes patients and without Diabetes. The paper itself has appropriate structure and it presents a good basis for much more detailed empirical research in future work. The paper is acceptable in current form, since it has been improved, according to the review suggestions.

Author Response

Dear Reviewer,

Thank you for your comments and acceptance on the previous revision. We appreciate your insightful comments on revising the content of the paper.

Reviewer 2 Report

The quality of the paper has not been improved sufficiently. My previous comments were addressed only in part. I do not think that the study was performed correctly. Specifically, game design must be based on a set of well established game design patterns, which is not the case with the one presented in this paper. Moreover, the development of a game in itself does not represent a scientific novelty. There are many such games available for smartphones now. The authors did not present a sufficient explanation of scientific novelty in their paper.

Author Response

Dear Reviewer,

Thank you for your comments. We appreciate your insightful comments on revising the content of the paper. I include the response as follow:

Point 1: Game design must be based on a set of well established game design patterns, which is not the case with the one presented in this paper.

Response: We tried our best to address all the comments. We might be misunderstood the comment in the previous revision. However, in this revision, we add one more paragraph on the gamification design pattern. We used game-design-based gamification guideline as our based to design the gamified application by analysing the suitability of our gamification design. This can be referred in line 328 – 369.

Point 2: The development of a game in itself does not represent a scientific novelty. There are many such games available for smartphones now. The authors did not present a sufficient explanation of scientific novelty in their paper.

Response: We add a paragraph of mobile apps review in the literature (refer line 179-219) that shows the mobile apps has a limited feature for catering the self-managements requirement for diabetes. One sentence also added in the introduction area (refer line 84-87).

We hoped the revised manuscript will be sufficient and accepted for the journal publication. We thank you for your continued interest in our research.